# Breeding Amaranth for Biomass: Evaluating Dry Matter Content and Biomass Potential in Early and Late Maturing Genotypes

**Ali Baturaygil [1]**, **Markus G. Stetter [2]** and **Karl Schmid [1,*]**

1   Institute of Plant Breeding, Seed Science and Population Genetics, University of Hohenheim, 70599 Stuttgart, Germany; ali.baturaygil@uni-hohenheim.de
2   Institute for Plant Sciences and Cluster of Excellence on Plant Sciences, University of Cologne, 50674 Cologne, Germany; m.stetter@uni-koeln.de
*   Correspondence: karl.schmid@uni-hohenheim.de

**Abstract:** Amaranth (*Amaranthus* spp.) is a promising biomass crop for silage and biogas production. Under long-day conditions, it exhibits prolonged vegetative growth. To evaluate the breeding potential of amaranth for biomass production, we characterized phenotypic variation in biomass yield components, quantitative genetic parameters, and the relationships between traits. We conducted field trials of 10 biomass-type genotypes exhibiting a 'giant' growth habit derived from spontaneous hybridization between genetically diverse parents, and used the variety "Bärnkrafft" as check. We observed two contrasting growth patterns: Bärnkrafft is a variety for grain production and was characterized by a short vegetative growth followed by a long seed ripening. In contrast, the biomass genotypes displayed a long vegetative growth followed by a short seed ripening. We observed strong correlations between dry matter content and stem diameter ($r = -0.78$, $p < 0.01$) and between plant height and biomass score ($r = 0.95$, $p < 0.001$). High values for broad-sense heritability of stem diameter ($H^2 = 0.88$) and plant height ($H^2 = 0.92$) suggest that the dry matter content and yield can be improved by indirect phenotypic selection. We hypothesize that selection for dry matter content and yield implies a trade-off between earliness and photoperiod sensitivity. Hence, dry matter content should be improved first by recurrent selection, which can be then combined with short-day genes to improve dry matter yield. Overall, this work provides an avenue to the breeding of biomass amaranth.

**Keywords:** amaranth; biomass; quantitative genetics; photoperiod sensitivity; dry matter yield

## 1. Introduction

The production of bioenergy is an important component in efforts to reduce dependence on fossil fuels. One way to produce bioenergy is the anaerobic digestion of plant material in bioreactors and a consecutive conversion of the resulting biogas into electricity and heat through a generator [1]. In Germany, biogas production from energy crops has grown rapidly with 9200 biogas plants that produce 4.2 GigaWatts as of 2016 [2]. Maize silage is the most popular biogas substrate in Germany, with a mass-based contribution of about 70% among energy crops [3]. Given that methane yield is mainly determined by dry matter yield, high dry matter yield is the primary breeding objective in biogas crops [4,5]. Maize has become the predominant biogas crop because it combines high dry matter yield and content [6]. However, potential negative impacts of maize monoculture, such as increased risk of soil erosion and a decrease in agrobiodiversity, create a demand for alternative energy crops [7,8].

Amaranth is a possible alternative bioenergy crop. The genus *Amaranthus* harbors more than 60 species, of which several species are cultivated as grain crops, leaf vegetables or ornamental plants [9]. In Central and South America, grain amaranth species are ancient

crops [10,11] that have been rediscovered in the last decades due to their favorable nutritional qualities [12]. However, most *Amaranthus* species are undomesticated and weeds in agricultural production areas [13,14]. Amaranths are C4 plants and therefore able to photosynthesize at high temperatures in conjunction with a higher water use efficiency than many C3 plants [15]. These characteristics facilitate the introduction of amaranth as a crop to dry and marginal zones [16,17]. In the temperate Central European climate amaranth showed better drought stress tolerance than maize in a comparison of both crops for biogas suitability [18]. In addition, amaranth can be used as forage crop [19–21].

*Amaranthus* species differ in their photoperiodic response [22], which can be utilized to develop varieties suitable for biomass production. Many amaranth species need a short daylight duration below 12 h for flowering induction. Among the three cultivated grain amaranths (*A. caudatus* L., *A. cruentus* L. *and A. hypochondriacus* L.), *A. caudatus*, requires less than eight hours [11,15,23]. In contrast, *A. cruentus* is the most photoperiod insensitive amaranth species [9]. Under long-day conditions of Central Europe, short-day crops delay flowering and prolong biomass accumulation [24]. Since most amaranths are short-day plants with an elongated vegetative growth under long-day environments, amaranth was considered as potential biomass and biogas crop for cultivation in Europe.

Amaranth is mainly self-pollinating with an out-crossing rate between 3–32% [25,26], which makes it an attractive species for plant breeding because it allows breeding methods used for both autogamous and allogamous crops. Even though up to 88% mid-parent biomass heterosis was observed in $F_1$ generation hybrids [27], an efficient large-scale method for hybrid seed production does currently not exist for amaranth. However, methods for experimental crosses have been successfully applied [28] and the existence of cytoplasmic male sterile (CMS) line and the restorer line (*A. hypochondriacus* L.) may allow a large-scale production of $F_1$ seeds and may serve to exploit biomass heterosis commercially in the future [29].

The biomass and biogas potential of amaranth was investigated in several studies [6,18,20,30–34]. Comparative studies evaluated whether amaranths are competitive with maize as a bioenergy crop and revealed that maize is superior to amaranth due to its high performance in both dry matter yield and content. Such an advantage of maize is expected because it has been improved by long-running commercial hybrid breeding programs that utilized heterosis [35–37]. As a consequence, a large number of high yielding biogas type maize varieties have been released. In contrast, breeding efforts in grain amaranths have been restricted to the selection of individual genotypes from landrace populations [38]. In vegetable amaranth, breeding efforts have been limited to the acclimatization of a small number of lines in India [39,40], but quantitative genetic parameters estimated in trials indicated a positive potential for future improvement of vegetable amaranth by breeding [41]. Overall, a lack of breeding activities likely contributes to the current position of amaranths as minor crop.

Although amaranth genebank accessions and landraces were evaluated for their suitability as biomass crops and for biogas production, no variety for biomass production was released to date [18]. The necessity of additional breeding efforts to improve amaranth as potential biogas crop was recognized [18]. So far, no study has investigated the plant breeding potential of amaranth as bioenergy crop, and estimated quantitative genetic parameters relevant for breeding. In this study, we evaluate the potential of breeding for biomass amaranth by: (1) characterizing phenotypic variation in biomass yield components, (2) determining the components of phenotypic variation and detecting correlations between traits, and (3) proposing a breeding strategy for amaranth with high dry matter yield. Our results suggest that amaranth could become a suitable addition to existing biomass crops by targeted breeding programs.

## 2. Materials and Methods

### 2.1. Plant Material

We focused on ten genotypes from our biomass amaranth breeding pool, whose ancestors include putative $F_1$ generation hybrids derived from spontaneous outcrossing events between Bärnkrafft, Puerto Moutt (*A. cruentus*), C6 (*A. caudatus*) and Pastewny (*A. hybridus*) that occurred during field trials in 2012, as well as multiple genebank accessions cultivated with these four genotypes. These species are diploid and their chromosome number is 2n = 32 [15]. We selected putative hybrid plants based on a giant growth habit and excessive plant height in 2013. Seeds of selected individuals were sown in a greenhouse after harvest to obtain $F_2$ generation plants. In the next growing season of 2014, we tested $F_2$ generation seeds originating from the 2013 field trial, as well as $F_3$ generation seeds produced in a greenhouse, and 120 genebank accessions from which individuals with gigantic growth habit in a field trial were selected. From all of these populations, we harvested seeds of putative biomass type individuals. Due to the large number of plants, they were not covered with bags, and for this reason, outcrossing was possible, which may contribute to heterogeneity in the next generation. Collected seeds were planted and the best ten individual plants were selected for intensive evaluation in repeated trials in 2015. Seeds of these individuals were used in our study. In addition, we included Bärnkrafft, which was the only amaranth variety registered in Germany at that time [42] as check variety. Bärnkrafft is a grain amaranth variety, and has a stable phenotype that has been selected for cultivation in Central European climates.

### 2.2. Experimental Design and Phenotypic Evaluation of Field Trials

In 2016, we tested the eleven genotypes at the Heidfeldhof agricultural station (48°43′07.3″ N 9°11′08.7″ E, 395 m a.s.l.) and the Eckartsweier agricultural station (48°32′52.4″ N 7°52′32.5″ E, 140 m a.s.l.) of the University of Hohenheim. These two locations differ in the distribution of precipitation and temperature during the vegetation period. Eckartsweier is a suitable growth environment for amaranth due to its high temperature and Heidfeldhof was used for receiving higher precipitation (Figure 1). The soil type was silty loam in both locations. The field trial had a randomized complete block design with three blocks per location that each contained the eleven genotypes in individual plots. Double-row plots had a length of 5 m with a distance of 0.75 m between rows. Plots within blocks were separated by 75 cm and between blocks by 1 m distance. Each experiment was surrounded by a check variety to prevent border effects. Sowing and thinning were conducted manually by leaving 10 cm distance between plants. The two experiments were planted on 4 May 2016 and 9 May 2016 and harvested on 12 October 2016 and 11 October 2016 in Heidfeldhof and Eckartsweier, respectively. Weed control was carried out manually and mechanically, and no irrigation or fertilization was applied. We recorded five biomass yield components: plant height (in cm), dry matter content (as the percentage) and stem diameter (in mm). Plant height was measured from the ground surface to the top of the inflorescence, and was recorded ten times during the growing season at Heidfeldhof and eight times at Eckartsweier. For this, 15 plants were randomly selected at the young plant stage approximately a month after the sowing in each plot and were labeled. All further measurements were taken from these individuals. For plant height, only the last measurements taken at harvest time in each location were used in the statistical analysis. Stem diameter was measured from 10 cm above the ground surface with a caliper, and recorded on the same 15 plants at harvest time. Dry matter content was estimated on five randomly selected individuals that were located in the inner part of a plot at harvest time. Harvested plants were cut above the ground surface and their roots were left in the soil. A sample from the fresh biomass from these five individuals was weighted (fresh weight). After drying in a ventilated oven at 110 °C for 72 h, samples were again weighed (dry weight). The ratio of dry weight to fresh weight was considered as dry matter content. We scored the plots visually for biomass and inflorescence volume at harvest time using the 1–9 scale, where

1 refers to the most inferior and 9 to the most superior performance based on the visual volume of the plots for the respective traits.

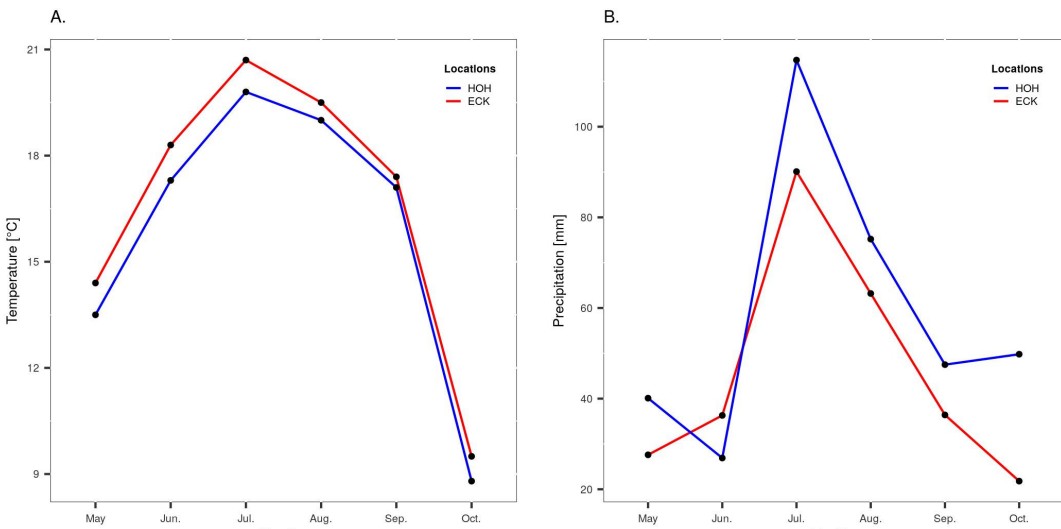

**Figure 1.** Monthly mean values of (**A**) temperature (°C) and (**B**) precipitation (mm) belong to the experimental locations between May–October 2016 (Agrarmeteorologie Baden-Württemberg, 2016).

### 2.3. Statistical Analysis

Location-specific and adjusted genotype means were estimated for each trait with a linear and a linear mixed model, respectively. In the estimation of adjusted genotype means, only genotype effect was taken as fixed and all other effects as random, whereas all effects in both the linear and linear mixed models were taken as random in the estimation of variance components. A Friedman test was used to estimate genotype means in the location-specific analyses of the score traits [43]. We used Kenward–Roger method to approximate degrees of freedom and standard errors in the linear mixed model analyses [44]. In the linear and linear mixed models, the significance of the genotype effect was evaluated with a type 3 test of fixed effects. In the linear model, linear mixed models, and Friedman test, pairwise comparisons of genotype means were conducted with Fisher's least significant difference (LSD) test at a significance level of 0.05.

We used the following linear mixed model:

$$y_{ijk} = \mu + \alpha_i + \beta_j + (\alpha\beta)_{ij} + b_{jk} + e_{ijk} \tag{1}$$

where $y_{ijk}$ is the observation value of response variable obtained from $i$-th genotype in $k$-th block in $j$-th location, $\mu$ is the overall mean, $\alpha_i$ the effect of $i$-th genotype, $\beta_j$ the effect of $j$-th location, $(\alpha\beta)_{ij}$ the interaction of the $i$-th genotype and $j$-th location, $b_{jk}$ the effect of $k$-th block nested within $j$-th location and $e_{ijk}$ the error associated with $y_{ijk}$.

The linear model was:

$$y_{ij} = \mu + b_j + \alpha_i + e_{ij} \tag{2}$$

where $y_{ij}$ is the observation value of response variable obtained from $i$-th genotype in $j$-th block, $\mu$ is the overall mean, $b_j$ is the effect of $j$-th block, $\alpha_i$ is the effect of $i$-th genotype and $e_{ij}$ the error associated with $y_{ij}$.

Broad-sense heritability $H^2$ was calculated as [45]

$$H^2 = (\sigma^2{}_g)/(\sigma^2{}_g + (\sigma^2{}_{gxe})/m + \sigma^2/rm) \tag{3}$$

and plot-based repeatability as [46]

$$w = (\sigma^2{}_g)/(\sigma^2{}_g + \sigma^2/r) \tag{4}$$

where $\sigma^2_g$ is the genetic variance, $\sigma^2_{gxe}$ is the genotype by environment interaction variance, $\sigma^2$ is the residual error variance, *m* is the number environments, and *r* is the number of replicates per environment.

Relationships between traits were studied based on adjusted genotype means using Pearson's correlation. We performed statistical analyses with linear mixed and linear models, and estimations of variance components using the MIXED procedure of SAS 9.4. [47]. The SAS/IM macro %MULT, developed by Piepho [48] was used for ensuring the robust notation of the significant differences among the pairwise comparisons in linear mixed and linear model analyses. The Friedman test was performed with the agricolae package [49] and the correlation analysis was performed using the GGally package of the R statistical environment [50].

## 3. Results

### 3.1. Differences between Biomass Types and Grain Types

For several quantitative and morphological traits we observed heterogeneity within lines due to residual genetic segregation (Figure 2). Lines with residual heterogeneity were most easily recognized in qualitative color traits such as inflorescence, leaf and stem color, whereas plant height and stem color were the quantitative traits with a high heterogeneity. To minimize the effect of variation within lines, we did not take averages for each plot, but collected observations from 15 randomly selected individuals per plot. The mean number of individuals recorded per plot was 13.70 (SD: 1.66) for plant height, and 13.85 (SD: 1.33) for stem diameter, respectively. Losses were mainly caused by an insufficient emergence of two or three genotypes at Heidfeldhof and rarely by plant lodging, which was equally distributed over plots.

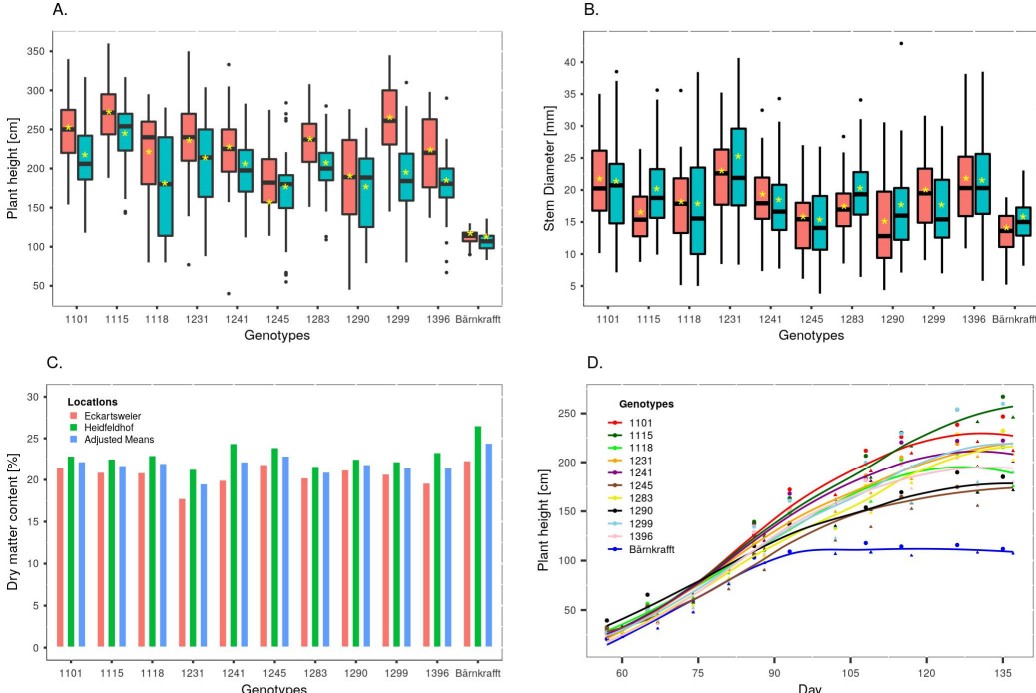

**Figure 2.** Box plots of genotypes for (**A**) plant height and (**B**) stem diameter, based on individual plant observations across three blocks per location. Red and turquoise colors indicate Eckartsweier and Heidfeldhof, respectively. Yellow asteriks indicate location-specific least square means of the genotypes. (**C**) Bar plot of dry matter content. Red and green colors indicate location-specific least square means of the genotypes from Eckartsweier and Heidfeldhof, respectively. Blue color indicates the adjusted least square means estimated over both locations. (**D**) Repeated measurements of plant height based on both locations. Dot and triangle point shapes represent Eckartsweier and Heidfeldhof, respectively. Each observation point indicates the mean value of a genotype based on the labeled individuals over three blocks, whereas the corresponding value of an observation point on the X axis indicates the observation day. The graph was generated using local weighted polynomial regression model [51].

The extent of heterogeneity varied among genotypes. The biomass genotypes differed in their heterogeneity, e.g., were more heterogeneous in quantitative traits such as plant height and stem diameter (Figure 2A,B) than Bärnkrafft, which was essentially uniform. Overall, the grain type variety Bärnkrafft differed strongly from the biomass genotypes in all three quantitative traits and was characterized by a lower plant height, smaller stem diameter, but higher dry matter content (Figure 2A–C). Bärnkrafft reached its final height after approximately 100 days in contrast to the other genotypes, which spent a longer time in the vegetative growth phase before switching to the generative growth phase (Figure 2D). Due to these differences, we conducted the following statistical analyses both with and without Bärnkrafft to evaluate the effect of this distinct variety on parameter estimates.

### 3.2. Variation in Biomass Yield Components

We estimated ranges for trait values based on adjusted genotype means of two locations (Table 1 and Supplementary Table S1). A wide phenotypic range was obtained for plant height (109–253 cm) and stem diameter (14–23 mm), and a narrower range for dry matter content (19 to 24%). The traits with scores (biomass, inflorescence volume) were normally distributed in the joint analysis and therefore directly used without data transformation. Biomass score was highly variable between genotypes (1.50–8.17), but the range was narrower for inflorescence volume score (3.67–7.67).

**Table 1.** Trait means, associated standard errors and ranges based on adjusted genotype means, mixed model analyses (Type 3 tests of fixed effects) with biomass yield components as dependent variables and genotype as fixed effect, and broad-sense heritability and plot-based repeatability values estimated in Heidfeldhof and Eckartsweier, including and excluding Bärnkrafft.

| | Type 3 Tests for Genotype Effect (Including Bärnkrafft) | | | | | Broad-Sense Heritability and Plot-Based Repeatability (Including Bärnkrafft) | | | | | | |
| --- | --- | --- | --- | --- | --- | --- | --- | --- | --- | --- | --- | --- |
| **Traits** | **Mean** | **±** | **SE** | **Range** | | **Num DF** | **Den DF** | ***F*-Value** | **Pr > F** | $H^2$ | $w_H$ | $w_E$ |
| Biomass score | 5.47 | ± | 0.52 | 1.50 | – 8.17 | 10 | 10 | 13.51 | 0.0002 *** | 0.93 | 0.91 | 0.97 |
| Inflorescence volume score | 5.41 | ± | 0.36 | 3.67 | – 7.67 | 10 | 10 | 6.08 | 0.0043 ** | 0.84 | 0.74 | 0.76 |
| Dry matter content (%) | 21.93 | ± | 0.38 | 19.59 | – 24.48 | 10 | 10 | 3.83 | 0.0227 * | 0.74 | 0.78 | 0.79 |
| Plant height (cm) | 199.90 | ± | 11.80 | 109.46 | – 253.36 | 10 | 10 | 11.87 | 0.0003 *** | 0.92 | 0.90 | 0.95 |
| Stem diameter (mm) | 18.18 | ± | 0.83 | 14.29 | – 23.46 | 10 | 10 | 8.23 | 0.0013 ** | 0.88 | 0.78 | 0.80 |
| | Type 3 Tests for Genotype Effect (Excluding Bärnkrafft) | | | | | Broad-Sense Heritability and Plot-Based Repeatability (Excluding Bärnkrafft) | | | | | | |
| **Traits** | **Mean** | **±** | **SE** | **Range** | | **Num DF** | **Den DF** | ***F*-Value** | **Pr > F** | $H^2$ | $w_H$ | $w_E$ |
| Biomass score | 5.87 | ± | 0.37 | 3.50 | – 8.17 | 9 | 9 | 5.94 | 0.007 ** | 0.83 | 0.82 | 0.93 |
| Inflorescence volume score | 5.27 | ± | 0.40 | 3.67 | – 7.67 | 9 | 9 | 5.60 | 0.0086 ** | 0.82 | 0.74 | 0.72 |
| Dry matter content (%) | 21.67 | ± | 0.28 | 19.59 | – 22.93 | 9 | 9 | 2.47 | 0.0966 ns | 0.60 | 0.44 | 0.75 |
| Plant height (cm) | 208.95 | ± | 8.37 | 161.36 | – 253.36 | 9 | 9 | 5.41 | 0.0096 ** | 0.82 | 0.75 | 0.89 |
| Stem diameter (mm) | 18.57 | ± | 0.81 | 14.91 | – 23.46 | 9 | 9 | 6.63 | 0.0047 ** | 0.85 | 0.75 | 0.74 |

*, **, *** significant at $p < 0.05$, $p < 0.01$ and $p < 0.001$, respectively, whereas ns shows non-significance. $w_H$ and $w_E$ reprsesents plot-based repeatability values estimated in Heidfeldhof and Eckartsweier, respectively.

In the joint analyses of both locations, Bärnkrafft reached the highest dry matter content in the experiment and it was different than other genotypes, except a single genotype ($p = 0.0227$, Supplementary Materials File S1), whereas we observed no difference among biomass genotypes when Bärnkrafft was excluded ($p = 0.0966$). In the location-specific analyses for dry matter content, Bärnkrafft was also different from the other genotypes (Supplementary Table S1). However, the biomass genotypes were not different from

each other in Heidfeldhof, but were different from each other in Eckartsweier, when we performed the analyses without Bärnkrafft (Supplementary Table S1).

Phenotypic trait means differed between the two locations. Larger trait means were estimated in Heidfeldhof for dry matter content and in Eckartsweier for plant height in location-specific analyses (Supplementary Table S1). Since the traits biomass and inflorescence volume score were not normally distributed in location-specific analyses, we performed a non-parametric Friedman test to compare genotype means and found genotypes to be different in both traits and in both locations (Supplementary Table S1). However, a Friedman test did not allow to estimate location-specific trait means, as it is a rank-based test and characterized by sample size. Therefore, we estimated median and interquartile range in these traits for the location-specific trait comparisons (Supplementary Table S1).

### 3.3. Quantitative Genetics Parameters and Relationships between Traits

We estimated two different broad-sense heritability and plot-based repeatability values for each quantitative trait both with and without Bärnkrafft (Table 1). Heritabilities were generally high, but showed lower values when Bärnkrafft was excluded. The lowest heritability was estimated for dry matter content ($H^2 = 0.74$). Similarly, large values were obtained for plot-based repeatability (Table 1). Like heritability estimates, repeatability values were lower when Bärnkrafft was excluded. Dry matter content showed the biggest difference in repeatability, with a value of 0.78 with and 0.44 without Bärnkrafft in the Heidfeldhof field trial.

We also correlated trait values based on the adjusted genotype means by including and excluding Bärnkrafft (Figure 3). The correlation between biomass score and plant height and between dry matter content and stem diameter were significant in both analyses. There was a strong positive correlation between biomass score and plant height with ($r = 0.95$, $p < 0.001$) and without Bärnkrafft ($r = 0.88$ $p < 0.001$). Dry matter content and stem diameter were negatively correlated with ($r = -0.78$, $p < 0.01$) and without Bärnkrafft ($r = -0.72$, $p < 0.05$). Dry matter content was negatively correlated with plant height ($r = -0.71$, $p < 0.05$) and biomass score ($r = -0.71$ $p < 0.05$). Finally, plant height showed a positive correlation with stem diameter ($r = 0.64$, $p < 0.05$) when Bärnkrafft was included.

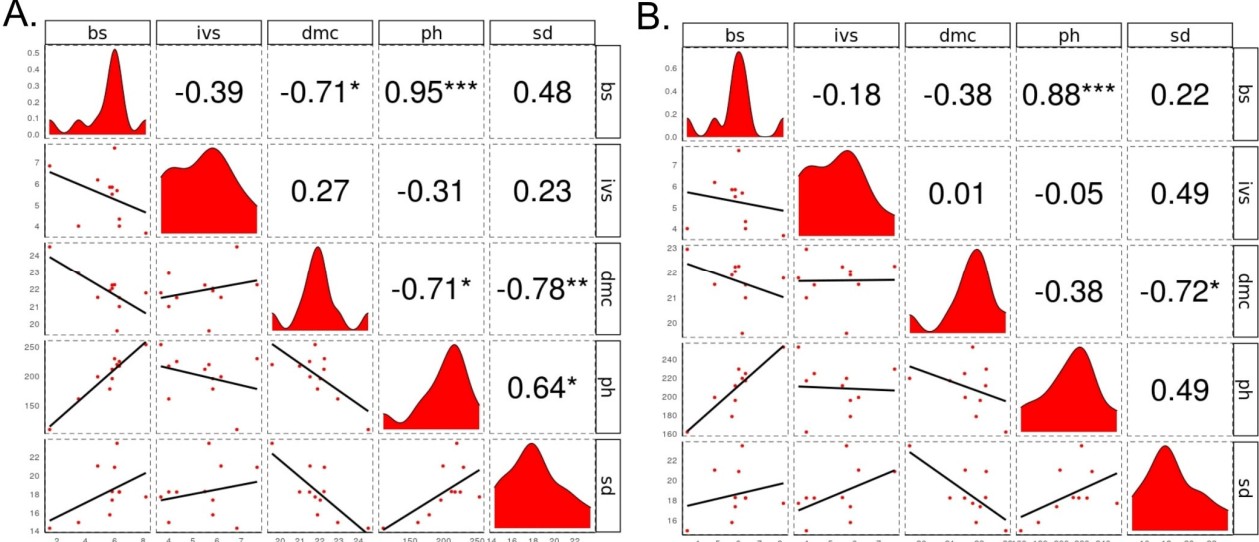

**Figure 3.** Pearson's correlation coefficients between biomass yield components, based on the adjusted genotype means over two locations. Bärnkrafft is (**A**) included and (**B**) excluded. Traits are bs, biomass score; ivs, inflorescence volume score; dmc, dry matter content; ph, plant height and sd, stem diameter. *, **, *** significant at $p < 0.05$, $p < 0.01$ and $p < 0.001$ respectively.

## 4. Discussion

### 4.1. Variation in Biomass Yield Components

We observed a strong difference in average dry matter content between the ten biomass lines (average 21.7 %) and the single grain type variety Bärnkrafft (24.5%, Supplementary Materials File S1). A higher dry matter content in Bärnkrafft is expected because of its earlier maturity. This difference explained also the significance in genotype effect for dry matter content in the joint analysis of all 11 genotypes and the non-significance in the 10 biomass genotypes. Dry matter content should be at least 28% for a satisfactory ensiling process in biogas production [5]. In our study, the grain type variety Bärnkrafft reached 26.6% dry matter content at the Heidfeldhof site and von Cossel et al. [18] reported up to 27.6%. These results suggest that amaranth has the potential to meet this requirement by further breeding. Furthermore, we note that amaranth has smaller particle sizes for seed and chaff than maize. Franco et al. [52] suggested that a smaller particle size improves methane yield production and a dry matter content threshold of 28% for maize may be be lower for amaranth.

We analyzed inflorescence volume score as an indirect measure of grain yield and found significant genotype effect when Bärnkrafft was both included and excluded. This significance presumably originated from variation in number of days required to reach physiological maturity among biomass genotypes. We also found genotypes to be different in the traits related to vegetative growth such as plant height and stem diameter probably because genotypes had a longer time until harvest to demonstrate their differences. Although we could not obtain dry matter yield values for comparison, a highly significant genotype effect in biomass score is a positive indicator for the improvement of dry matter yield (Table 1).

### 4.2. Trade-Off between Earliness and Photoperiod Sensitivity

We did not phenotype flowering time in our study due to residual segregation for this trait, however, the time point at which genotypes achieve a constant plant height can be used as a proxy for flowering time, when plants switch from vegetative to generative growth [53]. According to this definition, the difference in days for the beginning of flowering is quite large between Bärnkrafft and the biomass genotypes, but much smaller among the ten biomass genotypes (Figure 2). We suggest that the differences in dry matter content and plant height between Bärnkrafft and the biomass genotypes are mainly caused by variation in flowering time, as early flowering leads to a longer seed ripening phase and improved dry matter content, whereas late flowering due to short-day genes leads to longer vegetative growth and higher plant height.

Although the exact parents of biomass genotypes are unknown, they include *A. caudatus* and *A. cruentus* accessions, which are photoperiod sensitive and insensitive, respectively, that were involved in spontaneous crossing events [9,15,23]. Therefore, we hypothesize that short-day genes were introgressed into the biomass genotypes that lead to prolonged vegetative growth of the biomass genotypes by a combination of two factors: (I) the presence of short-day genes responsible for photoperiodic response and (II) long-day conditions during the growth phase that caused delayed flowering in the presence of short-day genes [22,54,55]. According to their genetic architecture of flowering time, individuals delayed or completely withheld flowering and continued to accumulate biomass throughout the cultivation period. Therefore, a widely known pattern in energy crops—delayed flowering leading to higher biomass yield—appears to hold true in amaranths as in other crops [22,24,55,56]. However, testing the effect of flowering time on biomass yield requires further analysis of segregating populations.

The composition of dry matter yield i.e., the contribution of grains to total dry matter yield is of crucial importance to secure sufficient dry matter content. In forage maize, cob to total dry matter yield ratio is around 50% [57], and the main breeding objective is digestibility, which is determined by dry matter content [4]. It promotes the use of earliness genes for higher grain yield and restricts the use of short-day genes. In contrast,

the main objective of biogas maize breeding is high dry matter yield, and the exploitation of short-day genes is more flexible, provided that 28% dry matter content is secured [4]. Similarly, in biomass sorghum, the use of short-day genes increases dry matter yield, but a low dry matter content requires to prioritize increased panicle contribution to total dry matter yield. Windpassinger et al. [58] propose the use of silage type sorghum for biogas production, whose ratio of panicle to total dry matter yield ranges between 40–50%, and therefore outperforms biomass type sorghum in methane yield and dry matter content.

Similar to the situation in these grass species, our study supports the hypothesis of a trade-off between earliness genes, for increased dry matter content, and short-day genes, for increased dry matter yield. We therefore propose that variation in flowering time is required to select for both earliness and photoperiod-sensitivity. Selection for a defined flowering time and the introduction of short-day genes may improve dry matter yield. More specifically, a flowering time interval within dry matter yield is maximized should be determined, since the inclusion of short day genes may likely influence such an interval across a set of environments. To achieve maximum dry matter yield, the optimal inflorescence to biomass ratio and its dependence on flowering time have to be evaluated. As a first step, selection for improved dry matter content should be prioritized and short-day genes can subsequently be used to improve dry matter yield, provided the requirement of a high dry matter content is fulfilled.

*4.3. Quantitative Genetics Parameters and Relationships between Traits*

In the first step, a suitable base population for dry matter content improvement can be generated by using photoperiod insensitive *A. cruentus* accessions, which have a panicle to total dry matter yield ratio of around 50% [30]. In the second step, high dry matter content can be combined with short-day genes to improve dry matter yield, by making crosses between photoperiod insensitive and sensitive genotypes. Accordingly, genotypes combining high dry matter content and prolonged vegetative growth can be selected from such populations with a large segregation variance. A similar approach succeeded in an energy maize breeding program in Germany, by combining photoperiod sensitivity genes from exotic Peruvian and Mexican populations, high grain yield potential from Italian populations and cold-tolerance genes from German populations [59]. In our study, stem diameter and plant height were highly heritable and also were strongly correlated to dry matter content and yield, respectively (Figure 3). Therefore, these traits seem promising to be used in an indirect phenotypic selection of the target traits. In addition, we observed dry matter content and inflorescence volume score to be nearly uncorrelated, particularly when Bärnkrafft is excluded. This can be explained with the low dry matter content variation of our biomass genotypes in contrast to high variation in inflorescence volume score that does not include a genotype with an outlier performance. Hence, future studies should represent grain type amaranths with more genotypes for re-examination of the selection efficiency of stem diameter and plant height and more accurate correlation estimates between grain yield components and dry matter content. Consistent with our study, moderate to strong positive correlations between plant height and dry matter yield ($r$ = 0.81 and 0.71) were also reported in biogas maize and sorghum, respectively [4,58].

The residual heterogeneity within plots was also a source of genetic variance, but is explained by the residual error term in the mixed model. This heterogeneity may cause an underestimation of genetic variance and an overestimation of the residual error term, which then results in an underestimation of broad-sense heritability and plot-based repeatability parameters. Since the residual error variance is larger than the genetic variance for dry matter content, heritability and repeatability may be underestimated for this trait, but its effect on genetic variance cannot be estimated with our design. Furthermore, the execution of multi-environment field trials across several years with a higher number of target environments would allow more accurate parameter estimations in future studies.

*4.4. Future Prospects*

　　　Future breeding efforts in biomass amaranth should primarily address the genetic characterization of flowering time and photoperiod sensitivity because of the role of variation in the trade-off of these two traits. Such a goal can be achieved because of the availability of a high-quality reference genome (*A. hypochondriacus* L.) and crossing methods to generate mapping populations [28,60–62]. Here, we focused on primary biomass traits, but traits like lignocellulose, sugar, protein, and lipid contents, as well as nutrients and trace elements can be alternative selection targets for an optimized biochemical composition of biomass amaranths [63]. Overall, the application of novel breeding methods such as genomic selection combined with speed breeding may rapidly improve the selection gain in the desired traits and promote the use of this minor crop as a resilient alternative to current biomass crops that is suitable for cultivation in marginal areas, and thereby reduces competition for food and feed.

**Supplementary Materials:** The following are available online at https://www.mdpi.com/article/10.3390/agronomy11050970/s1.

**Author Contributions:** Designed the experiment, A.B., M.G.S., and K.S.; developed the genotypes, M.G.S.; collected and analyzed the data, A.B.; wrote the manuscript, A.B. and K.S. All authors have read and agreed to the published version of the manuscript.

**Funding:** This work was supported by the F. W. Schnell endowed Professorship of the Stifterverband and the Hohenheim-Tübingen Regional Alliance of the Ministry of Science and Culture (MWK) of Baden Württemberg.

**Data Availability Statement:** The phenotypic data and the supplementary files are available from Figshare (10.6084/m9.figshare.c.5421621).

**Acknowledgments:** We thank Viola Abraham and the staff of the Hohenheim experimental stations for help with the field experiments and Hans-Peter Piepho for statistical advice.

**Conflicts of Interest:** The authors declare that the research was conducted in the absence of any commercial or financial relationships that could be construed as a potential conflict of interest.

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
