# Peer review of "Breeding Amaranth for Biomass: Evaluating Dry Matter Content and Biomass Potential in Early and Late Maturing Genotypes"

_agronomy, doi:10.3390/agronomy11050970_

Round 1

Reviewer 1 Report

The paper explores some biomass yield components variation across 10 short-day biomass flowering genotypes in comparison with a photoperiod insensitive grain variety.

A weakness of the study is the paucity of variables measured to characterize these materials. No phenology data, or biomass quality are provided  . The authors use proxies of agronomic traits such as biomass yield and seed yield by visually scoring biomass yield and inflorescence volume but made no attempt of calibration (ground truth validation) which would add methodological value to the work.  

I think the work provides some preliminary information on biomass potential of short-day flowering amaranth so deserves to be published.  I find the discussion redundant .  Basically the authors attribute all the differences between genotypes to the length of the vegetative growth without considering any differences in plant architecture or seed yield components and draw no conclusion on the specific material analysed .  Are there any lines that deserve future exploration ?  

I encourage the authors to be more succinct and to the point and discuss some practical implication of their work: what are the limits of this study, what would be the next step?

I have marked up the pdf with some suggestions reported below

Pag 3 line 137: Add soil texture , soil classification method , organic matter content and total N if available

Pag 4 line 157 : Please explain what do you mean by "ground tissue" , Could you explain the sampling procedure in detail

Pag 4 line 164  I suggest to replace this figure with a standard thermo-pluviometric diagram

Pag 5 line  206 Did the authors measure flowering time, if so why not reporting in results and discussion session? If these data are not available remove the comment from results and discussion

Pag 5 line  213- 216 I think this paragraph can be summarized in one line: "biomass genotypes were more heterogeneous than Barnkrafft"

Pag 5 line  218 – I would cancel the words “In addition”. This does not seem to me an additional factor: Barnkrafft is chracterized by a shorter vegetative cycle which explains why it has a lower plant height and a higher dry matter content (possibly due to leaf senescence and grain ripening)

Pag 5 line 220 : This is vague . Please quantify how long was the vegetative growth?  flowering stage was reached in how many days from sowing ?

Pag 6 Fig 2 : sort the genotypes in fig 2. A and B to check if there is any ranking across genotypes , I would take off the yellow asterisks which does not add much information but worsen readability . C. Use boxplot for dry matter content to be consistent with the information given in fig 2.A and B,

However if you compared means with LSD why not showing a bar plot with LSD bars , this would be informative of differences among genotypes than boxplots

Pag 7 Table 1 - Why the Numerator and denominator degrees of freedom are the same ?  Please explain how you calculated the degrees of freedom

Pag 7 line 257 -  I don’t see table S1 file S1 (???)What post-hoc test did you use after Friedman to compare the genotypes ?

Pag 8 line 283-284 - This is surprising to me I would expect that stem structural carbohydrates were positively correlated with dry matter content, please discuss your results

Pag 8 line 310 - This is vague, please explain how inflorescence morphology change during growth

Pag 9 lines 314-316 - Please explain . From correlation analysis I see biomass score and dry matter yield are almost uncorrelated

Pag 9 This sentence is not clear to me, I believe that you are discussing only one factor: the presence of short-day genes that by delaying flowering prolonged vegetative growth

Pag 10 Conclusion : How your results will contribute to increase the existing knowledge on amaranth breeding potential for biomass production ? what  is the novelty ? What are the main limits of the study and how can these limits be addressed by future research

Reviewer 2 Report

The authors have presented a manuscript, which evaluated earliness and photoperiod sensitivity in Amaranthus. Phenotypical parameters, dry matter in 11 amaranthus genotypes were analyzed under two different location. The manuscript presents interesting data concerning plant biomass and the bioenergy, but the document need to be improved. Following, I have included some comments aimed to enhance the paper:

  • Line 94, in the part of the objectives, it is better to no mentioned what is expected by the results.

  • L 96-111 in material and methods, this paragraph is from the Template archive and the journal author’s guide. Remove from the text.

  • Line 134-135, the authors indicated the location of the trial and refer to figure 1. It is better to referred to Figure 1 as data of temperature and precipitation for the two locations.

  • In this part of the manuscript, the authors must also explain why they choose these two locations and what difference there is between them.

  • Line 138, authors reported: three blocks per location, that each contained the eleven genotypes. While in the line 149, for this, 15 plants were randomly selected. The methodology is not clear; authors reported that each plot contained 11 genotyes. Improve explications concerning replicates and the genotypes sampling in the three Block.

  • Line 151, in material and methods, For plant height, only the last measurements taken at harvest time in each location were used in the statistical analysis. While in the Figure 2, Los authors presented the plant height in different date. Figur4 2(D) Repeated measurements of plant height.

  • Line 154-155, Dry matter content was estimated on five randomly selected individuals per plot at harvest time. It is not clear. Why authors have sampled only five of the 11 plants for each block? More explication for sampling.

  • In the Figure 2 C, present the standard error.

  • In Figure 2, the authors present the means of phenotypical measurements for the 11 genotypes, but it is better if they present also a statistical study as P values ​​to show if there is a significant difference between the 11 genotypes and in each location.

  • Line 323. We suggest that the differences in dry matter content and plant height between Bärnkrafft and the biomass genotypes are mainly caused by variation in flowering time. Justify it with some reference or another study.

  • Line 388-392, this paragraph is also from the Template archive and the journal author’s guide. Remove from the text. Authors must be very careful when presenting the document.

  • Authors presents study of earliness and photoperiod sensitivity. They associated these parameters with the height of plants. Why the dates and duration of flowering for each genotype have not been evaluated?

-          Finally, the topic of this manuscript is interesting; bioenergy is a very important theme. The earliness, photoperiod sensitivity, and dry matter are essential parameters, but authors must improve the presentation of their results and discussion.
